# Pathophysiology and Therapeutics of Thoracic Aortic Aneurysm in Marfan Syndrome

**DOI:** 10.3390/biom12010128

**Published:** 2022-01-14

**Authors:** Keiichi Asano, Anna Cantalupo, Lauriane Sedes, Francesco Ramirez

**Affiliations:** Department of Pharmacological Sciences, Icahn School of Medicine at Mount Sinai, New York, NY 10029, USA; keiichi.asano@mssm.edu (K.A.); anna.cantalupo@mssm.edu (A.C.); lauriane.sedes@mssm.edu (L.S.)

**Keywords:** angiotensin II, arterial disease, endothelial dysfunction, Marfan syndrome, NO signaling, oxidative stress, TGFβ, thoracic aortic aneurysm

## Abstract

About 20% of individuals afflicted with thoracic aortic disease have single-gene mutations that predispose the vessel to aneurysm formation and/or acute aortic dissection often without associated syndromic features. One widely studied exception is Marfan syndrome (MFS) in which mutations in the extracellular protein fibrillin-1 cause additional abnormalities in the heart, eyes, and skeleton. Mouse models of MFS have been instrumental in delineating major cellular and molecular determinants of thoracic aortic disease. In spite of research efforts, translating experimental findings from MFS mice into effective drug therapies for MFS patients remains an unfulfilled promise. Here, we describe a series of studies that have implicated endothelial dysfunction and improper angiotensin II and TGFβ signaling in driving thoracic aortic disease in MFS mice. We also discuss how these investigations have influenced the way we conceptualized possible new therapies to slow down or even halt aneurysm progression in this relatively common connective tissue disorder.

## 1. Introduction

Thoracic aortic aneurysm (TAA), leading to acute medial dissection (TAAD), is the most life-threatening manifestation of Marfan syndrome (MFS), a multi-system connective tissue disorder caused by mutations in fibrillin-1 [1]. This 350 KDa cysteine-rich glycoprotein is the main structural component of the 10 nm microfibrils that are found in virtually all extracellular matrices (ECM), either associated with elastin in the elastic fibers or as elastin-free macroaggregates [1,2]. Microfibril biogenesis is a complex and largely undefined process whereby fibrillin-1 monomers polymerize and grow linearly and laterally while incorporating other ECM proteins and interacting with cells via integrin and syndecan receptors [2]. Microfibrils also control cell behavior through sequestration and concentration of latent TGFβ complexes into the ECM [3]. Matrix metalloproteinases (MMPs) and integrins are the main molecules involved in the release and/or activation of ECM-bound latent TGFβ complexes during tissue formation, remodeling, and repair [3]. It follows that functional deficiency of such a critical component of this network of molecular interactions is expected to lead to a gradual-to-acute collapse of the tridimensional organization, homeostatic processes, and mechanical properties of aortic tissue.

Concentric layers of elastic fibers and vascular smooth muscle cells (SMCs), together with interconnecting microfibrils, form the lamellar units of the tunica media that impart elastic recoil and distribute hemodynamic tension throughout the vessel wall [4]. Microfibrils are also abundant in the ECM of the tunica intima, where they provide structural support to endothelium-mediated control of vascular tone by interacting with aortic endothelial cells (ECs) and linking them to the internal elastic lamina (IEL) [4]. Thoracic aortic disease in MFS is associated with endothelial dysfunction, SMC dedifferentiation and apoptosis, unproductive ECM remodeling, localized inflammatory infiltrates, and impaired mechanical properties [1]. Lifestyle modifications, regular echocardiographic assessment, treatment with anti-hypertensive drugs, and prophylactic surgery are currently used to manage TAAD in MFS patients [1]. However, there is still a need for more therapeutic options due to the heterogeneous and unpredictable progression of the disease, limited information about the efficacy of current β-blocker therapy, and significant morbidity/mortality risk of surgical repair.

Studies of mouse models of MFS have yielded several important new insights into the natural history, molecular mechanisms, and potential treatments of TAAD. First and foremost, characterization of mice genetically unable to produce fibrillin-1 (*Fbn1^−/−^* mice) revealed that, in agreement with the expression patterns of the respective genes, fibrillin-2 microfibrils support elastogenesis in the vessels of the developing embryo, and fibrillin-1 microfibrils fulfill the same function during the blood pressure surge period of early postnatal life [5]. In this respect, TAAD in MFS should be viewed as being a disease of failed tissue homeostasis. Additional studies of mice hypomorphic for fibrillin-1 or heterozygous for a structural fibrillin-1 mutation (*Fbn1^mgR/mgR^* mice and *Fbn1^C1041G/+^* mice, respectively) correlated aneurysm progression with improper stimulation of angiotensin II (AngII) type 1 receptor (At1r) signaling [6,7]. Promiscuous TGFβ signaling has also been reported in the dilating aorta of *Fbn1^mgR/mgR^* and *Fbn1^C1041G/+^* mice but with conflicting interpretations of whether this abnormality is a primary or secondary effector of disease [6,8]. More recent experimental findings from MFS mice have corroborated earlier clinical evidence of endothelial dysfunction affecting nitric oxide (NO) production [9,10]. Here, we briefly review the literature concerning the pathological significance and, implicitly, the therapeutic opportunities of endothelial dysfunction and dysregulated AngII and TGFβ signaling in the MFS aorta. Interested readers are referred to several excellent reviews discussing additional topics related to TAAD development in MFS and other genetic aortopathies.

## 2. AngII Signaling

The renin–angiotensin system (RAS) is a key regulator of circulatory homeostasis that influences vascular contractility and ECM production predominantly through activation of several At1r-mediated signaling pathways, including the Erk1 pathway [11,12]. The other major AngII receptor (At2r) is expressed during fetal life and is reactivated in pathological states to promote responses opposite to those of At1r, including stimulating vasodilation and NO production (Figure 1). Additionally, AngII and TGFβ pathways interact at many different levels with both synergistic and antagonistic effects on vascular homeostasis. RAS involvement in aneurysm development is largely based on several studies of different mouse models of arterial disease (including MFS mice) that have documented the therapeutic effects of inactivating the *At1r* gene or inhibiting the At1r receptor [11,12]. By contrast, information about a possible connection between RAS dysfunction and human TAA is very limited. Relevant to MFS, treatment with the angiotensin-converting enzyme inhibitor (ACEi) enalapril improved aortic distensibility and reduced aortic stiffness more effectively than β-blocker therapy, in a prospective study of 58 afflicted individuals [13]. Furthermore, treatment with the angiotensin receptor blocker (ARB) losartan was deemed TAA protective in a non-randomized, retrospective analysis of 18 pediatric MFS patients [14]. However, the majority of subsequent randomized trials performed on several large cohorts of adult and pediatric MFS patients showed no statistical differences between losartan and β-blocker therapy in slowing down TAA progression [1]. This finding may reflect the clinical, age, and gender heterogeneity of the human cohorts, as well as differences in losartan dosing and delivery between the mouse and human trials.

The mechanism of losartan action on TAA development in MFS mice remains to be fully clarified. Different investigators have argued that losartan prevents TAA formation either by blunting TGFβ hyperactivity or by enhancing NO signaling, in part as a result of shunting AngII signaling from type I to the type II receptor [15]. Sellers et al. [16] recently reported that TAA was still present and responsive to losartan in *Fbn1^C1041G/+^* mice lacking At1r activity, implying an off-target effect of the ARB rather than the postulated switch to protective At2r signaling. The additional finding that double treatment with losartan and L-NAME did not modify TAA pathology identified NO synthase (NOS) as the ARB’s off-target. However, reduced aneurysmal growth and attenuated ECM degeneration were reported by Chen et al. [17], in a study that employed the same *Fbn1^C1041G/+^*; *At1r^−/−^* model. This discrepancy may conceivably reflect the different imaging protocols used by the two laboratories to monitor aneurysm growth. Notably, death from ruptured TAAD was significantly delayed in *Fbn1^mgR/mgR^*; *At1r^−/−^* mice [18]. A comparable delay was also observed in *Fbn1^mgR/mgR^* mice with EC-specific, but not SMC-specific, *At1r* inactivation, thus supporting the notion that losartan mitigates arterial pathology largely by recalibrating the balance of endothelium-derived regulatory signals of vascular tone [18]. In this view, At2r-mediated AngII signaling represents a key aspect of restoring this homeostatic control. However, the aortic disease was unabated in *Fbn1^C1041G/+^* mice treated with the At2r agonist, compound C21 [19]. Several possible reasons were invoked to explain this negative outcome, including drug dosing, and the time in which treatment was initiated. As observed for experimentally induced aortic dilatation [20], it may be possible that double treatment with losartan and C21 could modify aortic pathology in MFS mice as well.

## 3. TGFβ Signaling

TGFβ proteins (TGFβs 1–3) are secreted as a tripartite complex (large latent complex (LLC)) made of the dimeric cytokine non-covalently associated with the pro-peptides (small latent complex (SLC)), which confer latency and are, in turn, bound to a latent TGFβ-binding protein (LTBP-1, -3 or -4) [3]. After secretion, the LLC is sequestered into ECM via LTBP interaction with fibrillin, fibronectin, and collagen assemblies. This last step provides a spatially organized reservoir of latent TGFβ molecules that can be rapidly mobilized through the action of integrins and/or proteases, leading to receptor-mediated activation of the canonical Smad2/Smad3 pathway (Figure 2). Enhanced TGFβ signaling in the medial layer of the MFS aorta was originally identified in *Fbn1^C1041G/+^* mice and indirectly supported by evidence of constitutively high levels of phosphorylated (p)-Smad2 in human TAA specimens and isolated SMC cultures [6,21,22]. Prevention of TAA development and normalization of aortic p-Smad2 levels by treatment with either a neutralizing TGFβ antibody or the ARB losartan suggested a primary disease contribution of an overactive At1r/TGFβ axis [6]. This conclusion was indirectly supported by the discovery that mutations in distinct components of the canonical TGFβ signaling cascade cause aortic disease [23]. Although an important contributor to arterial disease, TGFβ’s role as a primary trigger of TAA formation in MFS was more recently refined by a series of studies using MFS mice and mice deficient for TGFβ signaling in postnatal SMCs.

In line with TGFβ’s protective role in AngII-induced aneurysms, SMC-specific disruption of the type II TGFβ receptor (Tgfbr2) in newborn wild-type mice (*TBR2^SMΔ^* mice) or *Fbn1^C1041G/+^* mice promoted TAAD and exacerbated TAA, respectively [24,25]. Likewise, systemic TGFβ neutralization initiated at postnatal day 16 (P16) accelerated TAAD progression in *Fbn1^mgR/mgR^* mice [8]. The fact that P16-initiated losartan treatment delayed TAAD progression in these MFS mice decoupled enhanced TGFβ signaling from heightened AT1r activity, a conclusion further strengthened by the distinct stages of maximal accumulation of surrogate molecular markers of the two pathways (p-Smad2 at P60 and p-Erk1 at P16, respectively) [8]. The late surge of TGFβ signaling was correlated with driving unproductive remodeling of a fibrillin-1 deficient aortic matrix [8]. TGFβ dimorphism during TAAD progression (protective from birth to around P45 pathogenic afterward) resembles the cytokine’s role as a tumor suppressor in the early stages of malignancy and as a pro-metastatic factor at later stages of the disease [26]. The formation of stage-specific LTBP/TGFβ complexes may be one of the mechanisms contextualizing the nature of the TGFβ signal. Such an argument has been invoked to explain TAAD prevention in *Fbn1^mgR/mgR^*; *Ltbp3^−/−^* mice [27]. A similar phenomenon was observed in *TBR2^SMΔ^* mice in that the incidence of dissection was inversely proportional to the age of tamoxifen-induced, Cre-mediated *Tgfbr2* inactivation [24]. *TBR2^SMΔ^* mice were also shown to display endothelial dysfunction associated with NO deficiency and aortic hypercontractility, indicating that there are TGFβ-dependent regulatory signals that originate from medial cells to instruct intimal cell behavior [28]. TGFβ-based therapies should, therefore, consider the timing of treatment, i.e., late rather than early in disease progression, and the specificity of the targeted pathways, i.e., those driving unproductive ECM remodeling rather than maintaining vascular tone.

## 4. Endothelial Dysfunction

Emerging evidence suggests that endothelial dysfunction is an early trigger of arterial disease in MFS. ECs regulate multiple physiological functions, including SMC relaxation/contraction and the inflammatory response, largely through NO production [9,12,29]. This potent short-lived vasodilator is produced by three NOS isoforms: endothelial (eNOS), inducible (iNOS), and neuronal (nNOS). Resting EC generates NO through receptor-dependent agonists (such as acetylcholine, Ach) and mechanical stimuli. Once produced, NO diffuses to the adjacent SMC where it activates soluble guanylate cyclase (sGC)/cyclic GMP (cGMP)/cGMP-dependent protein kinase I (PKG-I) signaling pathway to promote vasorelaxation (Figure 3). NO bioavailability is affected by oxidative stress (reactive oxygen species (ROS) augmentation), which promotes the production of the highly reactive and cytotoxic peroxynitrite (ONOO^−^) at the expense of protective NO, shifting its role from vasorelaxant to pro-oxidant and inducing ROS-mediated cellular damage. Moreover, low NO cellular level may also inhibit mitochondrial K_ATP_ channel (mitoK_ATP_) opening, trigger the opening of permeability transition pores (PTPs), and further increase the oxidative stress caused by mitochondrial ROS release, which propagates an autoregulatory loop of ROS-induced ROS formation. Notably, studies of *Fbn1^C1041G/+^* mice and aortic SMC cultures from MFS patients associated mitochondrial dysfunction with aneurysmal disease [30,31]. Both AngII and TGFβ pathways interact functionally with NO and ROS [9,12,32]. AngII regulates NO production in ECs and SMCs, whereas NO downregulates At1r activity. In vitro evidence suggests that HUVEC cultures subjected to shear stress upregulate TGFβ production with subsequent NO activation [33]. ROS mediate TGFβ signals that stimulate SMC dedifferentiation and apoptosis and MMP activation, whereas AngII stimulates ROS production by influencing assembly and expression of NADPH oxidase (NOX) complexes [9,12,19]. Additionally, EC-derived ROS increases aortic wall susceptibility to Ang II-mediated dissection [34].

Clinical studies of MFS patients documented lower vasodilatory response to flow and Ach-mediated stimulation of NO-dependent relaxation of the brachial artery, supporting a peripheral endothelial abnormality associated with low NO production and aneurysm formation [35,36,37]. Endothelial dysfunction in MFS mice was first identified by van Breemen’s laboratory, in two studies that demonstrated loss of basal NO production due to reduced eNOS activity and elevated oxidative stress associated with greater iNOS activity [38,39]. Subsequent investigations confirmed these early findings and provided a more detailed picture of endothelial dysfunction in MFS mice. Augmented ROS levels were identified in aortic tissues and SMC cultures from MFS patients in association with increased NOX4 activity [40]. Activation of a novel TGFβ/NOX4 axis in the endothelium was suggested to be responsible for eNOS uncoupling and arterial disease in *Fbn1^C1041G/+^* mice [41]. Attenuated TAA pathology in *Fbn1^C1041G/+^*; *Nox4^−/−^* mice provided genetic proof of ROS involvement [40]. Redondo’s laboratory reported high iNOS expression levels in the aortic media of both MFS patients and *Fbn1^C1041G/+^* mice [42,43]. Since iNOS is upregulated during oxidative stress and inflammation, this observation suggests the presence of a local pro-inflammatory state in the MFS aorta. Furthermore, they showed that iNOS-derived NO drives aneurysm formation in *Fbn1^C1041G/+^* mice through upregulation of sGC/cGMP/PKG-I axis, and that pharmacological and/or genetic inhibition of different components of this signaling axis prevented TAA development [43]. Inhibition and even reversion of arterial disease in *Fbn1^C1041G/+^* mice by lentiviral mediated *Prkg1* silencing greatly contrast the relatively modest attenuation of TAA pathology observed in *Fbn1^C1041G/+^*; *Nox4^−/−^* mice [40,43]. This difference may reflect the relative contributions of ROS augmentation and iNOS hyperactivity to TAA pathology or the genetic strategies used to inactivate the *Nox4* and *Prkg1* genes. The reproducibility of jugular vein injection to deliver high-titer lentiviral vectors to the aorta is also controversial [44,45]. Another puzzling finding is how forced expression of lentivirus transduced genes in *Fbn1^C1041G/+^* mice could completely reverse (as opposed to slowing down) the degenerative process of a fibrillin-1 deficient ECM [42,43]. Although the aforementioned pathological findings have yet to be mechanistically connected to one another, we can at least speculate that endothelial dysfunction in *Fbn1^C1041G/+^* mice might be accounted for by impaired eNOS-derived NO production associated with augmented ROS that stimulates iNOS-derived NO production in the media. An additional limitation of this speculative disease model is the lack of information regarding whether fibrillin-1 deficiency triggers endothelial dysfunction by affecting IEL integrity, altering the flow, promoting wall stress, or a combination of all these mechanisms. With these important caveats in mind, current evidence suggests that either restoring eNOS-mediated NO production or blunting excessive iNOS activity may in principle represent two suitable therapeutic strategies.

## 5. Conclusions

Our understanding of TAAD pathophysiology and therapeutics in MFS has evolved since the original identification of a pathogenic losartan-responsive AT1r/TFGβ axis to the current more complex disease model requiring a multidrug strategy to slow down or even halt aneurysm progression. This evolution has been fueled, in part, by the use of more rigorous approaches to characterize arterial disease in mice. As future investigations will focus on integrating mechanistically disparate findings related to endothelial dysfunction and improper AngII and TGFβ signaling, there will be an increasing need to expand and standardize experimental protocols across various laboratories, so as to compare with more confidence data from different drug treatments and mutant mice. As already noted, the exclusive reliance on histomorphology and echocardiography to monitor experimentally induced changes in TAA pathology is the main limitation of using *Fbn1^C1041G/+^* mice [46]. An additional experimental limitation of this MFS model is that TAA develops slowly and rarely dissects. The advantage of the other MFS model (*Fbn1^mgR/mgR^* mice) is that aneurysm progression is significantly faster and dissection is fully penetrant, thus allowing one to monitor possible drug/gene-induced modifications of TAAD by also recording relative changes in mouse survival [47]. Notably, a reliable new protocol has been published that avoids underestimating aortic dimensions by using in situ imaging of OCT-injected specimens [48]. Study designs should take into account experimental variability due to TAAD sexual dimorphism or the method (genetic vs. pharmacological intervention) used to target a particular signaling pathway [49,50]. They should also include ex vivo vasomotor and biomechanical analyses to more appropriately assess the impact of genetic and pharmacological interventions on aortic tissue function [51,52]. Along the same lines, the emerging importance of endothelial dysfunction in aortic disease will require the use of more appropriate methodologies to correctly and reproducibly measure oxidative stress and secondary products [53]. It is also likely that system biology approaches will be employed more extensively to delineate single-cell transcriptomics, predict probable interactions among different cell types, identify druggable nodes within disease-related molecular networks, repurpose FDA-approved drugs, and use human datasets to authenticate computational predictions derived from mouse aortic transcriptomes [54].

## Figures and Tables

**Figure 1 biomolecules-12-00128-f001:**
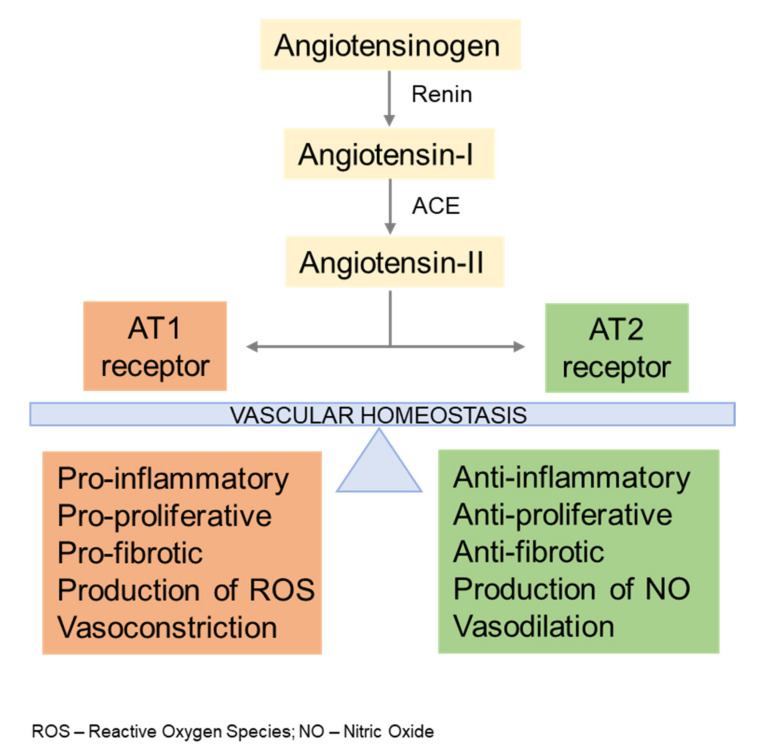
The renin–angiotensin system (RAS) in vascular homeostasis. The RAS system consists of a two-step enzymatic cascade. Renin, the first and rate-limiting enzyme, converts angiotensinogen into angiotensin I, which is, in turn, cleaved by angiotensin-converting enzyme (ACE) into angiotensin II that controls vascular functions by binding to and activating the AT1 and AT2 receptors.

**Figure 2 biomolecules-12-00128-f002:**
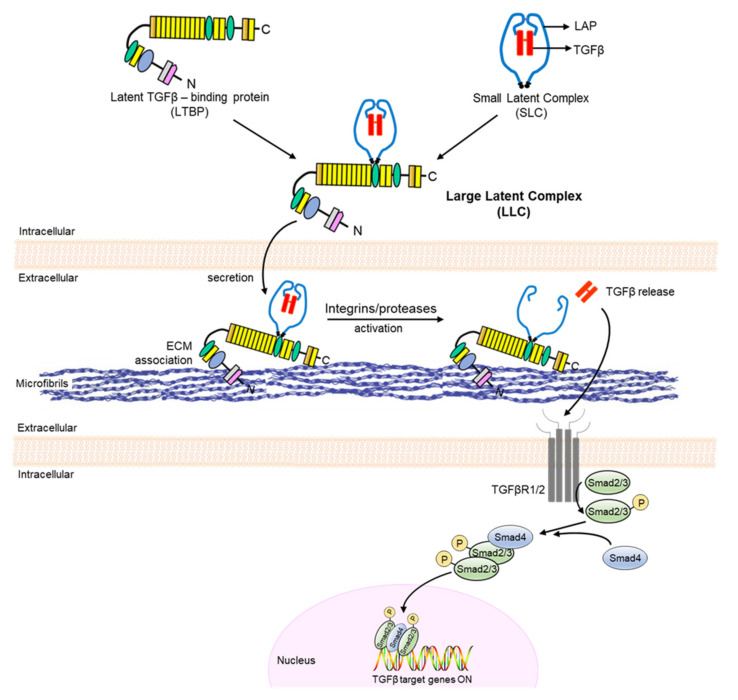
TGFβ latent complex, activation, and signaling. TGFβ is produced in the small latent complex that binds intracellularly with a latent TGFβ-binding protein to form the large latent complex, which associates extracellularly with fibrillin-1 microfibrils. After integrin/protease-mediated activation and release, TGFβ signals through receptor-mediated activation of the canonical Smad2/3 pathway.

**Figure 3 biomolecules-12-00128-f003:**
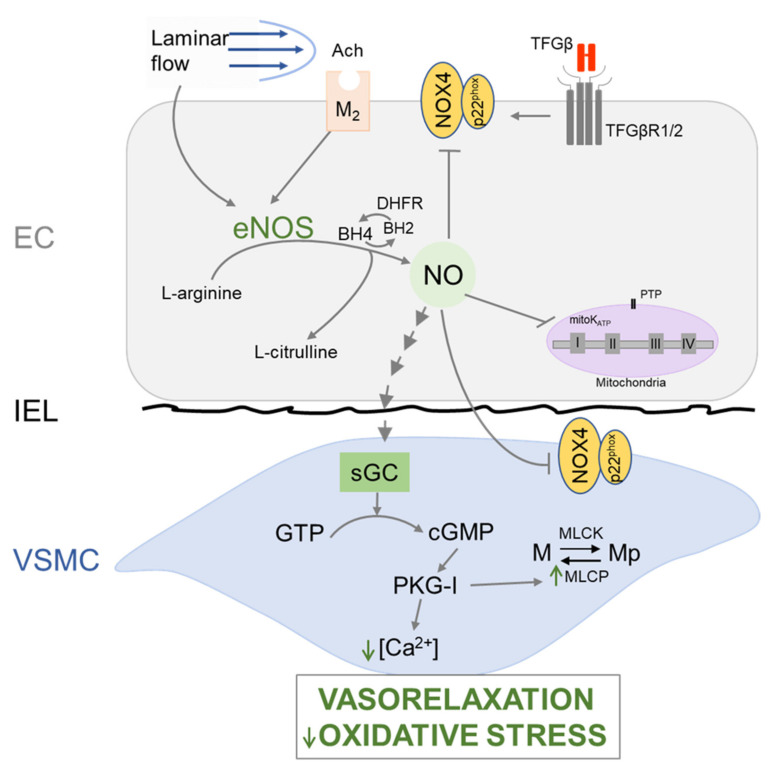
Nitric oxide (NO)/cGMP/PKG-I signaling pathway. In response to humoral or mechanical stimuli (e.g., Ach or shear stress), NO is synthesized in EC from L-arginine by activated form of eNOS. NO diffuses to neighboring SMC, where it activates soluble guanylate cyclase (sGC), which subsequently increases the intracellular production of cGMP from GTP. cGMP activates cGMP-dependent protein kinase I (PKG-I), which reduces intracellular Ca^2+^ concentration, and activates myosin light chain phosphatase (MLCP) to dephosphorylate myosin light chain (M), leading to SMC vasorelaxation. EC-derived NO can buffer ROS and prevent oxidative stress by inhibiting NOX activation in EC and SMC. It may also inhibit mitochondrial K_ATP_ channel (mitoK_ATP_) opening, and consequently PTP opening, thus preventing the oxidative stress caused by mitochondrial ROS release.

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
