# Peer review of "Pathophysiology and Therapeutics of Thoracic Aortic Aneurysm in Marfan Syndrome"

_biomolecules, 2022, doi:10.3390/biom12010128_

Round 1
Reviewer 1 Report
Asano and colleagues well reviewed the state of the art on therapeutic options for Marfan syndrome mice with thoracic aortic aneurysm, suggesting new interesting research avenues and novel potential biological item as therapeutic targets for the human pathological context.
The review is well written, the take-home massage is clear and is pleasant as well as interesting to read.
I would like to suggest just few minor revisions:
- Line 162: please, change "pathologienic" with a correct word.
- Line 199: please, correct "EC-derived-ROS increases aortic wall susceptibility" sentence.
- Figure 3:
- since the review's message came through loud and it is also clear how the AngII and TGFb mechanisms involve, at last, also the NO-dependent sGC/cGMP/PKG-I axis, it could be interesting to split the Figure 3 in two parts, one dedicated to the mechanisms in physiological conditions and the other to the pathological counterpart.
- please, double check "TFGβ" and "TFGβR1/2" writings.
- change PTTP with PTP in the figure capture.
Reviewer 2 Report
In general, the manuscript is a very well written and a sharp minireview, placing many controversies into perspective and creating opportunities to use the data to our advantage. However, I have a few suggestions still.
Line 86: “Furthermore, treatment with the angiotensin receptor blocker (ARB) losartan was deemed TAA protective in a non-randomized retrospective analysis of 18 pediatric MFS patients [14]. However, subsequent randomized trials of large cohorts of adult and pediatric MFS patients showed no significant differences between losartan and b-blocker therapy in slowing TAA progression [1]. This negative finding might….” I would formulate this paragraph a bit more neutral, because apart from these 18 patients, there is data out there in much larger cohorts where an encouraging effect of ARBs is observed (for example: Lancet. 2019 Dec 21;394(10216):2263-2270. doi: 10.1016/S0140-6736(19)32518-8. and Eur Heart J. 2020 Nov 14;41(43):4181-4187. doi: 10.1093/eurheartj/ehaa377), so I think the final verdict is not out there. Based on the murine data more was expected of it, but perhaps this was too optimistic considering the wide variety of Marfan patients (FBN1 mutations, modifier genes and behavior) compared to a uniform and well controlled Marfan mouse model. ARBs are now considered a good alternative next to beta blockers, so this is not a negative finding. However, the beta blockers or ARBs do not halt disease progression, thus improving therapeutics is essential for Marfan patients.
Line 177: “Emerging evidence suggests that endothelial dysfunction is an early trigger of arterial disease in MFS.” I think that the finding that impaired flow-mediated dilation (as readout of endothelial dysfunction) correlates with aortic dilation in patients with Marfan syndrome is very interesting and could be mentioned in this respect to human EC indications in MFS. (Heart Vessels. 2014 Jul;29(4):478-85. doi: 10.1007/s00380-013-0393-3.)
Line 189: “Moreover, low NO cellular level may also inhibit mitochondrial K-ATP channel (mitoK-ATP) opening, trigger the opening of permeability transition pores (PTTP), and further increase the oxidative stress caused by mitochondrial ROS release, which propagates an autoregulatory loop of ROS-induced ROS formation.” This also supports the mitochondrial dysfunction that is observed in Marfan human cells (Cardiovasc Res. 2018 Nov 1;114(13):1776-1793. doi: 10.1093/cvr/cvy150.) and is causal to aortopathy in the Marfan mice (Circulation. 2021 May 25;143(21):2091-2109. doi: 10.1161/CIRCULATIONAHA.120.051171 and ), which could be mentioned as pathological mechanism, either by ROS or change in metabolism to influence SMC and EC health.
Line 231: “Inhibition and even reversion of arterial disease in Fbn1C1039G/+ mice by lentiviral mediated Prkg1 silencing greatly contrast the relatively modest TAA attenuation seen in Fbn1C1039G/+;Nox4-/- mice [36,39]. This difference may reflect the relative contributions of ROS augmentation and iNOS hyperactivity to TAA pathology or the genetic strategies used to inactivate the Nox4 and Prkg1 genes.” Alternatively, it seems that the calcium homeostasis in the SMC may play a key role downstream of Prkg1. The disturbed calcium handling is shown by a number of people among which in the iPSC derived Marfan SMC (Nat Genet. 2017 Jan;49(1):97-109. doi: 10.1038/ng.3723.). Overactivation of Prkg1 by itself induces aneurysm formation in humans (Arterioscler Thromb Vasc Biol. 2017 Jan;37(1):26-34. doi: 10.1161/ATVBAHA.116.303229. and/or J Biol Chem. 2020 Jul 24;295(30):10394-10405. doi: 10.1074/jbc.RA119.010984.). This could be mentioned in this discussion (and in the scheme; see next comment).
It would be visually attractive to also have a figure showing all the things that potentially go wrong as described in the text and may be at play in the Marfan (aneurysm) aorta. Perhaps as Figure 3B? And then it becomes more evident which pathological pathways could be blocked in Marfan to decrease aortopathy.
Minor:
…permeability transition pores (PTTP)…; why 2xT? In Fig 3 it is indicated on the mito membrane as PTP. Which one is it? PTTP or PTP.
First the Fbn1 C1041G/+ mice are named by their actual mutation in mice, but in the second part of the manuscript the same mice are named by their old name of the human cysteine location; Fbn1 C1039G/+. I prefer consistency, thus all the novel nomenclature of Fbn1 C1041G/+.
In ref 38: metalloprotease Admats1, should be Adamts1.
